# Association of Whole-Heart Myocardial Mechanics by Transthoracic Echocardiography with Presence of Late Gadolinium Enhancement by CMR in Non-Ischemic Dilated Cardiomyopathy

**DOI:** 10.3390/jcm11226607

**Published:** 2022-11-08

**Authors:** Karolina Mėlinytė-Ankudavičė, Paulius Bučius, Vaida Mizarienė, Tomas Lapinskas, Gintarė Šakalytė, Jurgita Plisienė, Renaldas Jurkevičius

**Affiliations:** 1Department of Cardiology, Medical Academy, Lithuanian University of Health Sciences, LT-50161 Kaunas, Lithuania; 2Institute of Cardiology, Lithuanian University of Health Sciences, LT-50162 Kaunas, Lithuania

**Keywords:** myocardial fibrosis, non-ischemic dilated cardiomyopathy, 2D echocardiography, cardiac magnetic resonance

## Abstract

Background: In patients with non-ischemic dilated cardiomyopathy (NIDCM), myocardial fibrosis (MF) is related to adverse cardiovascular outcomes. The purpose of this study was to evaluate the potential relationship between the myocardial mechanics of different chambers of the heart and the presence of MF and to determine the accuracy of the whole-heart myocardial strain parameters to predict MF in patients with NIDCM. Methods: We studied 101 patients (64% male; 50 ± 11 years) with a first-time diagnosis of NIDCM who were referred for a clinical cardiovascular magnetic resonance (CMR) and speckle tracking 2D echocardiography examination. We analyzed MF by late gadolinium enhancement (LGE), and the whole-heart myocardial mechanics were assessed by speckle tracking. The presence of MF was related to worse strain parameters in both ventricles and atria. The strongest correlations were found between MF and left ventricle (LV) global longitudinal strain (GLS) (r = −0.586, *p* < 0.001), global circumferential strain (GCS) (r = −0.609, *p* < 0.001), LV ejection fraction (LVEF) (r = 0.662, *p* < 0.001), and left atrial strain during the reservoir phase (LASr) (r = 0.588, *p* < 0.001). However, the binary logistic regression analysis revealed that only LV GLS, GCS, and LASr were independently associated with the presence of MF (area under the curves of 0.84, 0.85, and 0.64, respectively). None of the echocardiographic parameters correlated with fibrosis localization. Conclusions: In NIDCM patients, MF is correlated with reduced mechanical parameters in both ventricles and atria. LV GLS, LASr, and LV GCS are the most accurate 2D echocardiography predictive factors for the presence of MF.

## 1. Introduction

Heart failure (HF) is a major cause of morbidity and mortality worldwide and is expected to increase due to the aging population. Cardiomyopathies are a group of structural and functional disorders that often lead to HF [1]. NIDCM is a severe form of primary myocardial disease associated with progressive HF and a poor prognosis. Recent studies recognize that pathogenetic mechanisms in NIDCM stimulate various fibrogenesis pathways [2]. Cardiac fibrosis is an important subject and should be evaluated quickly in each case of NIDCM because it is associated with adverse clinical outcomes, including cardiac arrhythmias, HF severity, higher rates of cardiac transplantation, and implantable devices [1]. It is also a useful tool for risk stratification in patients with early-stage dilated cardiomyopathy (DCM) [3,4]. LGE imaging is a powerful, noninvasive method for the estimation of the extent of cardiac fibrosis. Recently, multiple studies have suggested that the presence and extent of LGE have an incremental prognostic value over LVEF in NIDCM [5,6]. However, CMR imaging has limited availability; therefore, it is not always performed and rarely repeated in NIDCM patients as the follow-up is based on echocardiography. Advanced echocardiographic techniques, such as speckle tracking echocardiography (STE), allow the detection of structural cardiac abnormalities beyond LVEF and have been shown to correlate with the presence of MF in various clinical scenarios [7] including NIDCM [8]. However, the data to describe the relationships between the presence of MF and the whole-heart myocardial mechanics are lacking. In this context, we sought to evaluate the relationships between the mechanical properties of all cardiac chambers, as assessed by STE, and the presence of LGE on CMR in a prospective cohort of NIDCM patients.

## 2. Materials and Methods

### 2.1. Study Population

This study population consisted of 101 patients with NIDCM treated at the Hospital of Lithuanian University of Health Sciences Kaunas Clinics between January 2019 and January 2022. The NIDCM diagnosis was defined according to the World Health Organization criteria and the latest European Society of Cardiology (ESC) proposal [9,10]. The exclusion criteria were:ischemic coronary disease (ICD). All patients underwent invasive coronary angiography. ICD was defined as a history of myocardial infarction, revascularization, and the presence of epicardial coronary artery diameter stenosis > 50%;primary valvular heart disease;chronic kidney disease (eGFR < 30 mL/min/1.73 m^2^);under the age of 18;a poor echocardiographic or CMR image quality;inflammatory myocardial disease;previous pulmonary embolism;peripartum cardiomyopathy;toxic damage (alcohol, drugs).

All patients underwent a comprehensive clinical evaluation, laboratory test assessment, electrocardiogram (ECG), Holter monitoring (for the detection of rhythm disorders—ventricular tachycardia/ventricular fibrillation, ventricular premature beats, etc.), 2D transthoracic echocardiography, and CMR with LGE. Ethical approval was obtained for this study by the Kaunas Regional Biomedical Research Ethics Committee, and all participants gave written informed consent prior to enrollment.

### 2.2. 2D Echocardiographic Analysis 

#### 2.2.1. Standard Echocardiographic Parameters

2D echocardiography was performed using commercial ultrasound systems (Philips EPIQ 7) according to a prespecified protocol with recommendations by the European Association of Cardiovascular Imaging [11]. All images were stored digitally, and analysis was performed offline (TomTec Imaging Systems, Unterschleissheim, Germany) from archived cases in Digital Imaging and Communications in Medicine (DICOM) format. The patients were studied in the left lateral decubitus position by the same echocardiographer. 

For 2D cardiac morphometric measurements, end-diastole was defined as the cardiac cycle frame when the LV or right ventricle (RV) internal diameter was the largest and end-systole as the frame when the LV or RV cavity was the smallest. Left atrial volume (LAV) and right atrial volume (RAV) were calculated using the apical biplane area-length method and were indexed to body surface area. LV end-diastolic volume (LVEDV), LV end-systolic volume (LVESV), and LVEF were measured from the apical four- and two-chamber views and calculated by Simpson’s biplane method, following manual delineation of the endocardial border in the largest (end-diastolic) and smallest (end-systolic) boundaries. LVEF (%) was estimated by the following formula: (LVEDV−LVESV)/LVEDV × 100% [11].

#### 2.2.2. Speckle-Tracking Echocardiography

The EACVI/ASE/Industry Task Force consensus document was used to standardize left atrial (LA), right ventricular, and right atrial (RA) myocardial deformation parameters [12]. 

RV longitudinal strain (RVLS) values were assessed from the modified RV-focused apical 4-chamber view. Global RV longitudinal strain was evaluated by averaging peak strain values from six segments: three from the RV free wall and three from the interventricular septum. RV free wall longitudinal strain was averaged from the free wall segments only [12] (Figure 1).

The single apical four-chamber view was used to automatically assess the values for LA strain during the reservoir (LASr), conduit (LAScd), and contraction (LASct) phases. To obtain the LA strain values from an optimized (i.e., nonforeshortened) apical four-chamber view, we used the Task Force recommendations [12]. The zero-baseline for LA and RA strain curves was set at ventricular end-diastole using R-R electrocardiogram (ECG) gating as recommended. The RA endocardial border was manually traced in a four-chamber view, thus delineating a region of interest (ROI) composed of six segments. The reservoir (RASr), conduit (RAScd), and contraction (RASct) phases to evaluate the RA function were analyzed [12] (Figure 2). 

For the assessment of GLS, apical four-chamber, two-chamber, and long-axis views were acquired. LV circumferential and radial strains were measured by endocardial tracing in the basal, middle, and apical levels of LV short-axis views. The GLS, GCS, and global radial strain (GRS) were defined as the average peak strain values automatically generated from the 16 segmental strain curves by the software [13] (Figure 3).

### 2.3. Reproducibility of Myocardial Strain Measurements

Twenty patients were randomly selected, and a Bland-Altman analysis was performed to evaluate the intraobserver and interobserver variability, which shows a good agreement with a small bias of 0.5 ± 2.7% and 0.7 ± 3.9%, respectively.

### 2.4. CMR Imaging Protocol and Analysis

The CMR scans were performed on a 3T magnetic resonance imaging scanner with an 18-channel cardiac coil (MAGNETOM Skyra, Siemens Healthcare, Erlangen, Germany). Images were acquired during an expiratory breath hold with ECG gating. CINE images were acquired using standard balanced steady-state free precession sequences in three long axes (2-, 3-, and 4-chamber views) and an appropriate amount of short axis images (8 mm slices with a 2 mm inter-slice gap). LGE images were acquired in identical positions following an intravenous injection of gadobutrol (0.1 mmol/kg). LGE images were visually analyzed by an experienced reader for the presence and type (ischemic vs. non-ischemic) of enhancement (Figure 4). Subjects with a non-ischemic pattern of LGE were included in this study (non-ischemic myocardial injury can be observed at the epicardium, in the mid-wall, or at insertion points).

### 2.5. Statistical Analysis

The data were analyzed using SPSS version 22 (IBM, Chicago, IL, USA). We considered a *p*-value of <0.05 to be statistically significant. Continuous variables were expressed as means ± standard deviations (SD). Categorical variables were presented as absolute numbers and percentages and were compared using the Chi-square test. The Kolmogorov–Smirnov test was used to assess the normal distribution of the data. The student’s t-test was used to compare normally distributed variables, and the Mann–Whitney U-test was used for abnormally distributed variables among groups. We divided our study population into two groups according to the presence of MF (fibrosis positive and fibrosis negative). If one variable was categorical and one was continuous, the point-biserial coefficient of correlation R^2^ was calculated. A binary logistic regression analysis was used to determine the predictors of MF. The receiver operating characteristic (ROC) curve was plotted to evaluate the predictive value of echocardiographic strains to detect MF. The cut-off value of the predictive model was defined as the point that yielded the maximum value of the sum of sensitivity and specificity.

## 3. Results

The baseline demographic and clinical characteristics of the patients and the data from cardiac 2D echocardiography are shown in Table 1 and Table 2. MF was detected in 51 (50.5%) patients. The mean age in both groups was 50 years (*p* = 0.832). Patients with MF had higher rates of ventricular arrhythmias (ventricular tachycardia or fibrillation) compared to those without (52% vs. 14.3%, respectively, *p* < 0.001) or with severe HF (NYHA class III or IV, 68% vs. 46.9%, respectively, *p* = 0.045). 

The patients with MF had more dilated LV (difference between groups with *p* < 0.001). Moreover, the presence of MF was associated with a lower LVEF (*p* < 0.001) and more reduced LV strain values (*p* < 0.001). RV fractional area change was better in patients without LGE (30.1% vs. 33.5%, *p* = 0.002). RV-free wall longitudinal strain and GLS were more reduced in the group with the presence of MF. Both atria were more dilated (only volume index) and had the worse function in the group with MF (LAV index difference between groups with *p* = 0.021; RAV index difference between groups with *p* = 0.031; all strain parameters between groups with *p* < 0.001). 

The relationships between myocardial mechanics and MF are shown in Table 3. The strongest correlation was found between the presence of MF and mechanical strain parameters of the left side of the heart, including LV and LA strain values (*p* < 0.001). A weak correlation was found with RV strain parameters, RA reservoir strain (r = 0.379, *p* < 0.001), and RA contractile strain (r= −0.369, *p* < 0.001). A moderate correlation was observed between MF and RA conduit strains (r = −0.420, *p* < 0.001). 

Binary logistic regression analysis showed that only LV GLS, LV GCS, and LASr were independently associated with the presence of MF (Table 4). 

To evaluate the performance of the independent echocardiographic variables, we compared the area under the curve (AUC) of the ROC curves of LV GCS (Figure 5A), LV GLS (Figure 5B), and LASr (Figure 5C). Overall, GCS (AUC, 0.85; 95% CI, 0.77–0.92; *p* < 0.001), LV GLS (AUC, 0.84; 95% CI, 0.76–0.92; *p* < 0.001), and LASr (AUC, 0.64; 95% CI, 0.53–0.75; *p* = 0.013) were the strongest 2D echocardiographic predictors in the prognostication of MF. A cut-off value of −14.4 of LV GCS could prognosticate the presence of MF with a sensitivity of 100% and specificity of 85%. A LV GLS cut-off value of −9.5% could prognosticate the presence of MF with a sensitivity of 100% and specificity of 83%. The optimal cut-off value of LASr for the detection of MF was 11.7%, with a sensitivity of 86% and specificity of 82%.

## 4. Discussion

The findings of this study can be summarized as follows: (1) In patients with NIDCM, myocardial mechanics are more impaired in patients with the presence of MF; (2) LV GCS, LV GLS, and LASr are the best 2D echocardiographic deformation parameters to prognosticate the presence of MF.

Cardiac fibrosis occurs early in the progression of NIDCM, increasing cardiac rigidity, decreasing myocardial performance, and enhancing the risk of sudden cardiac death and life-threatening arrhythmias. Cardiac fibrogenesis is a multistep process initiated in response to pro-inflammatory stimuli and cytokines. Many cells, such as macrophages, monocytes, T lymphocytes, mast cells, and endothelial cells, are involved in the development of MF [2]. 

Early studies using LGE by CMR to assess for the presence of MF have reported that one-third to two-thirds of DCM patients have focal MF [14,15]. In recent years, the field has shifted towards finding a relationship between the presence and extent of MF and adverse clinical outcomes, such as cardiac arrhythmias, disease severity, higher rates of cardiac transplantation, implantable devices, and all-cause mortality [4,14,16]. However, there is a lack of data about the relation between the whole-heart myocardial deformation parameters and the presence or absence of MF.

We did not find an association between the clinical data of patients, such as age and gender, and the presence of MF. The data are lacking on the influence of gender or age on the prevalence and extent of LGE in NIDCM in other studies also. Some authors have reported that only the male sex predicts the presence of MF [4], and these findings are confirmed in patients with NIDCM [4,17]. In our study group, there were statistically significantly more men than women (men vs. women: 71 (64%) and 30 (27%), respectively), but there was no difference in the presence of MF. Our results show that more ventricular tachycardia or fibrillation episodes were detected in patients with MF, and these findings confirm the fact that cardiac fibrogenesis is associated with major arrhythmic events [18,19]. 

Previous data have suggested a strong correlation between MF and the worsening of HF [4,20]. Our results corroborated these findings. For a long time, LVEF was the main parameter on which diagnostic and treatment decisions were based; however, it was later shown that GLS is more sensitive in finding early LV dysfunction and has the strongest accuracy for detecting LV fibrosis [21]. As confirmed earlier, the impaired myocardial contractility affects the subendocardial layers first, and the fibers from this layer are responsible for longitudinal deformation [22]. In our cohort, LVEF had a moderate correlation with the presence of MF and was better in patients without MF. We have also found that LV GLS is one of the best 2D echocardiographic prognostic factors to predict MF. Interestingly, other authors present the results with no similar relationship between LV function and the extent of MF. It could be explained by the fact that only patients with end-stage HF were investigated [23]. It is relevant to notice that MF and changes in myocardial geometry in NIDCM mainly occurred in the middle layer of the myocardium. These abnormalities strengthen the probability of ventricular arrhythmias and promote the reduction of myocardial deformation values, especially GCS [24]. In contrast to other studies, which reported the most important value of GLS [25], our results showed that decreased LV GCS was a prognostic marker of MF in NIDCM as well. 

Lisi et al. reported that RV-free wall longitudinal strain was independently associated with RV fibrosis and was the main determinant of MF [23]. However, this study had a limited cohort of 27 patients and was comprised only of patients with end-stage HF (LVEF ≤ 25%). Our results have shown a statistically significant reduction of right heart strain values in the group with MF. However, we did not find any strain value of the RV or RA to be an independent parameter to predict the presence of MF. 

Despite the well-established focus on LV deformation parameters as predictors of outcomes in many cardiac diseases, the prognostic value of LA function has been recently gaining attention as well (involving heart failure, hypertension, renal failure, diabetes, etc.). Some studies have revealed that the LA reservoir strain is a sensitive prognostic marker [26,27]. LA longitudinal strain is a prognostic marker of atrial fibrosis predisposing for atrial fibrillation [28]. In our study, LA reservoir strain was the prognostic marker to detect MF and the presence of MF, was associated with lower LA strain values.

Unlike LASr, RA strain values showed only a weak correlation with the presence of MF. This could be explained by the fact that NIDCM more often affects the left-sided chambers of the heart. Additionally, RA strain is technically difficult to measure due to the difficulty in contouring thin walls and unusual chamber geometry [29]. 

Our data support that LV GLS, LV GCS, and LASr are the most important prognostic factors for the presence of MF.

### Limitations

There are several limitations to our study. First, this is a single-center cohort study with a small sample size. Secondly, our cohort mainly consisted of white men, which is not necessarily representative of all patients with NIDCM. Further studies in multi-center cohorts with extended follow-up are needed to verify the results of this study. Thirdly, we did not quantify the extent of LGE in this cohort, as we thought that the cohort was too small for further differentiation to yield results from which reliable conclusions could be drawn. Additionally, we only used 2D echocardiographic strain parameters for the assessment of myocardial mechanics. Finally, no follow-up data were available; thus, the prognostic value of our findings is not known.

## 5. Conclusions

In patients with NIDCM, MF is correlated with reduced mechanical parameters in both ventricles and atria. However, it is more closely related to the reduced strain values of LV and LA. LV GLS, GCS, and LARs are the strongest predictive factors for the prediction of MF. Clinicians should consider the routine use of these markers for early suspicion of MF and potential referral to CMR. 

## Figures and Tables

**Figure 1 jcm-11-06607-f001:**
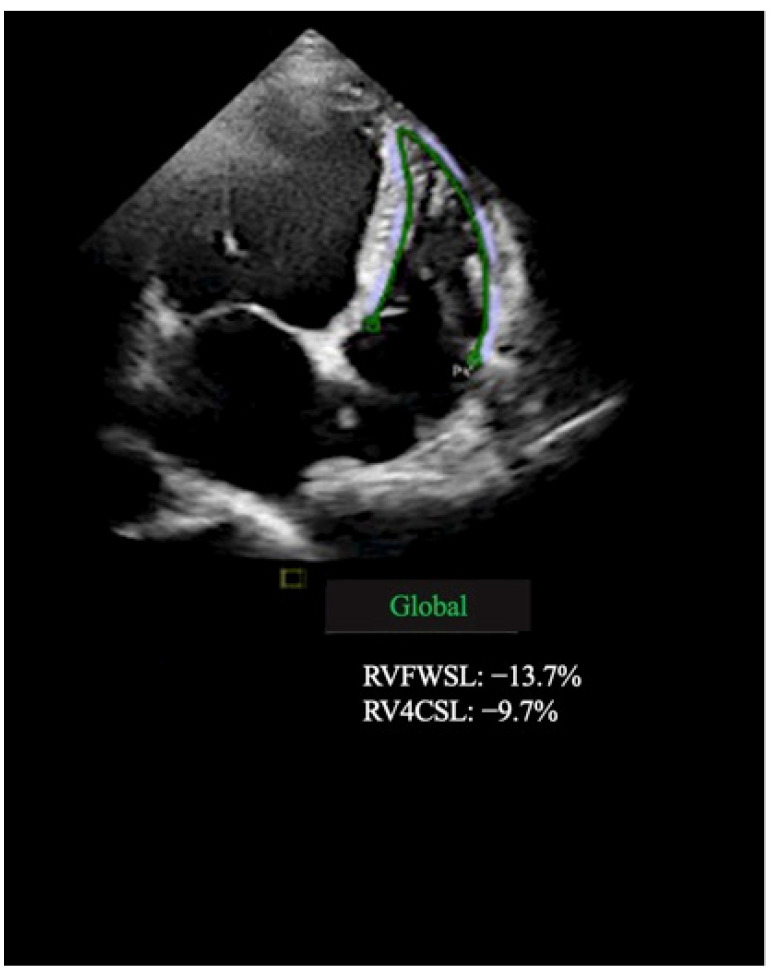
Right ventricular speckle tracking analysis shows decreased strain values using an apical four-chamber view with automated delineation of endocardial surface (green line).

**Figure 2 jcm-11-06607-f002:**
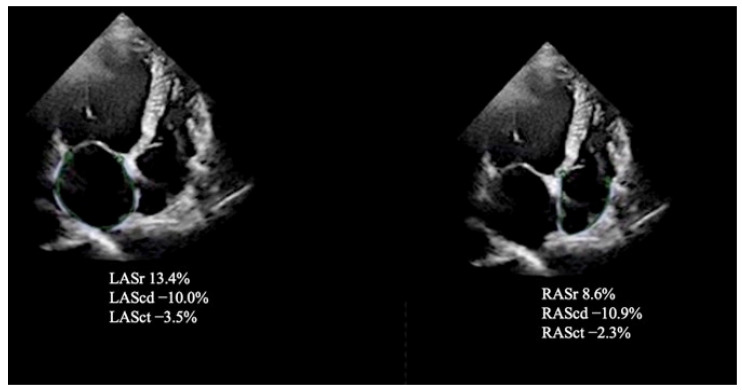
Left (on the left side) and right (on the right side) atrial speckle tracking analysis shows decreased strain values using apical four-chamber views with automated delineation of endocardial surface.

**Figure 3 jcm-11-06607-f003:**
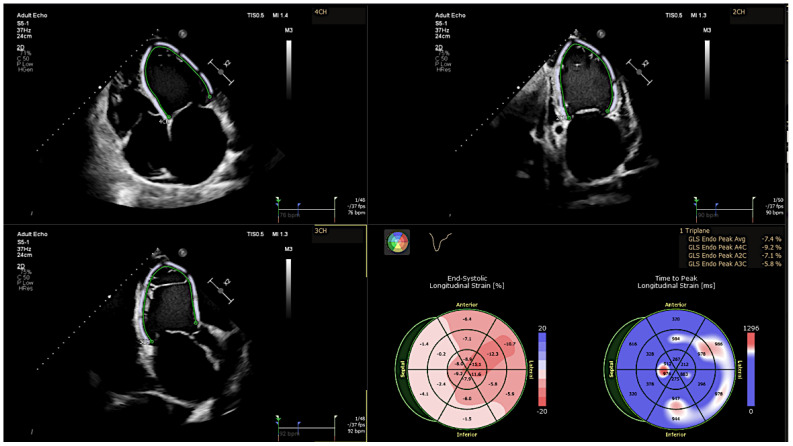
Left ventricular speckle tracking analysis shows decreased GLS strain values using apical four-, three-, and two-chamber views.

**Figure 4 jcm-11-06607-f004:**
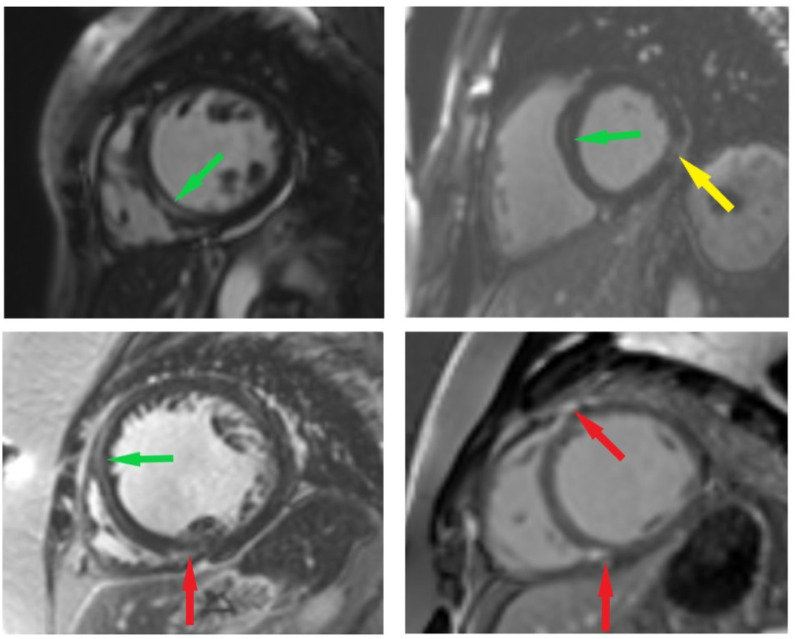
Short axis LGE images in four different patients with dilated cardiomyopathy. Three different types of replacement fibrosis can be seen: mid-wall (depicted by green arrows); RV insertion point (depicted by red arrows); and subepicardial (depicted by yellow arrows).

**Figure 5 jcm-11-06607-f005:**
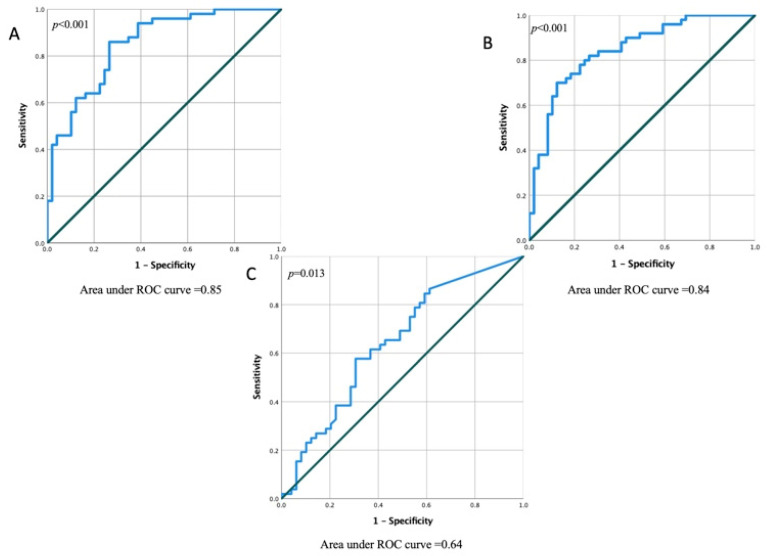
Area under the curve of the model in predicting MF probability in patients with NIDCM. (**A**) A ROC curve for GCS; (**B**) A ROC curve for LV GLS; (**C**) A ROC curve for LASr. Blue line—GCS (**A**), LV GLS (**B**), LASr (**C**); green line—reference line.

**Table 1 jcm-11-06607-t001:** Demographic and clinical characteristics in NIDCM patients with and without fibrosis.

Variables	Fibrosis Positive (LGE+) n = 51	Fibrosis Negative (LGE) n = 50	*p* Value (Fibrosis Positive vs. Fibrosis Negative)
Age, y	50.7 ± 10.8	50.3 ± 10.9	0.832
Males, n (%)	37 (74)	33 (67.3)	0.467
BSA, m^2^	2.0 ± 0.2	2.0 ± 0.2	0.725
Heart rate, beat/min	82.3 ± 18.2	77.3 ± 16.01	0.160
Systolic blood pressure, mmHg	124.1 ± 12.8	126.6 ± 13.9	0.353
Dyslipidemia, n (%)	21 (42)	19 (38.8)	0.838
Smoking, (%)	23 (46)	20 (40.8)	0.686
VT/VF, n (%)	26 (52)	7 (14.3)	<0.001
LBBB, n (%)	26 (52)	18 (36.7)	0.158
AF/AFL, n (%)	22 (44)	19 (38.8)	0.685
HF symptoms > 3 months, n (%)	33 (66)	39 (79.6)	0.176
QRS duration, ms	122.6 ± 30.6	119.9 ± 29.1	0.657
NYHA class III-IV, n (%)	34 (68)	23 (46.9)	0.045
BNP, ng/L	1540.2 ± 651.6	980.1 ± 714.7	0.106

BSA—body surface area; VT/VF—ventricular tachycardia/ventricular fibrillation; LBBB—left bundle branch block; AF/AFL—atrial fibrillation/atrial flutter; HF—heart failure; NYHA—New York Heart Association; BNP—brain natriuretic peptide; LGE—late gadolinium enhancement.

**Table 2 jcm-11-06607-t002:** 2D echocardiographic parameters in NIDCM patients with and without fibrosis.

Variables	Fibrosis Positive (LGE+) n = 51	Fibrosis Negative (LGE) n = 50	*p* Value (Fibrosis Positive vs. Fibrosis Negative)
IVS, mm	9.5 ± 1.3	9.9 ± 1.1	0.118
PW, mm	9.6 ± 1.4	9.8 ± 1.1	0.484
LVESD, mm	58.8 ± 7.5	53.1 ± 7.5	<0.001
LVESDi, mm/m^2^	29.1 ± 4.6	26.0 ± 3.9	<0.001
LVEDD, mm	66.7 ± 6.8	62.7 ± 5.0	0.001
LVEDDi, mm/m^2^	33.1 ± 1	30.7 ± 3.5	0.003
LA, mm	38.7 ± 4.6	37.8 ± 4.3	0.989
LAAi, mm/m^2^	19.8 ± 3.2	19.2 ± 2.8	0.988
LVEDV, mL	256.9 ± 80.6	209.7 ± 56.4	<0.001
LVEDVi, mL/m^2^	126.8 ± 41.1	103.1 ± 30.3	<0.001
LVESV, mL	188.9 ± 70.9	135.7 ± 45.3	<0.001
LVESVi, mL/m^2^	96.8 ± 41.8	68.2 ± 25.2	<0.001
GLS, %	−7.1 ± 2.1	−10.3 ± 2.5	<0.001
GCS, %	−11.1 ± 3.4	−16.9 ± 4.2	<0.001
GRS, %	15.5 ± 7.8	25.7 ± 7.8	<0.001
LVEF, %	22.1 ± 6.0	33.6 ± 7.3	<0.001
RV free wall LS, %	−17.3 ± 2.9	−19.0 ± 2.9	0.008
RV GLS, %	−9.5 ± 3.5	−13.2 ± 4.2	0.007
RV FAC, %	30.1 ± 5.9	33.5 ± 5.1	0.002
LAV, mL	129.4 ± 62.3	104.2 ± 63.5	0.05
LAVi, mL/m^2^	63.4 ± 32.1	49.8 ± 25.1	0.021
LAScd, %	−10.7 ± 3.2	−16.0 ± 4.1	<0.001
LASr, %	19.1 ± 6.1	27.4 ± 6.0	<0.001
LASct, %	−8.1 ± 2.6	−11.6 ± 4.9	<0.001
RAV, mL	84.3 ± 24.1	75.9 ± 24.4	0.091
RAVi, mL/m^2^	41.3 ± 10.8	36.8 ± 9.4	0.031
RAScd, %	−13.5 ± 5.1	−18.0 ± 4.8	<0.001
RASr, %	26.9 ± 6.4	31.6 ± 5.1	<0.001
RASct, %	−10.4 ± 6.9	−14.6 ± 3.1	<0.001

IVS—interventricular septum; PW—posterior wall; LVESD—left ventricle end-systolic diameter; LVESDi—left ventricle end-systolic diameter index; LVEDD—left ventricle end-diastolic diameter; LVEDDi—left ventricle end-diastolic diameter index; LA—left atrial; LAAi—left atrial area index; LVEDV—left ventricle end-diastolic volume; LVEDVi—left ventricle end-diastolic volume index; LVESV—left ventricle end-systolic volume; LVESVi—left ventricle end-systolic volume index; GLS—global longitudinal strain; GCS—global circumferential strain; GRS—global radial strain; LVEF—left ventricular ejection fraction; RV—right ventricle; LS—longitudinal strain; FAC—fractional area change; LAV—left atrial volume; LAVi—left atrial volume index; RAV—right atrial volume; RAVi—right atrial volume index; LASr—left atrial strain during reservoir phase; LAScd—left atrial strain during conduit phase; LASct—left atrial strain during contraction phase; RAScd—right atrial strain during conduit phase; RASct—right atrial strain during contraction phase; RASr—right atrial strain during reservoir phase.

**Table 3 jcm-11-06607-t003:** Correlation analysis of myocardial fibrosis and whole-heart myocardial mechanics.

	Fibrosis	
	rs	*p*
GLS, %	−0.586	<0.001
GCS, %	−0.609	<0.001
GRS, %	0.553	<0.001
LVEF, %	0.662	<0.001
RV free wall LS, %	−0.244	0.015
RV FAC, %	0.274	0.006
RV GLS, %	0.282	0.005
LAScd, %	−0.570	0.001
LASr, %	0.588	0.001
LASct, %	−0.409	0.001
RAScd, %	−0.420	0.001
RASr, %	0.379	0.001
RASct, %	−0.369	0.001

GLS—global longitudinal strain; GCS—global circumferential strain; GRS—global radial strain; LVEF—left ventricular ejection fraction; RV—right ventricle; LS—longitudinal strain; FAC—fractional area change; LASr—left atrial strain during reservoir phase; LAScd—left atrial strain during conduit phase; LASct—left atrial strain during contraction phase; RAScd—right atrial strain during conduit phase; RASct—right atrial strain during contraction phase; RASr—right atrial strain during reservoir phase.

**Table 4 jcm-11-06607-t004:** Binary logistic regression analysis for the variables associated with a presence of myocardial fibrosis.

Parameter	Exp (B)	95% CI	*p*
LV GLS, %	0.637	0.494–0.821	<0.001
LV GCS, %	0.828	0.715–0.958	0.011
LV GRS, %	1.002	0.911–1.103	0.058
LVEF, %	1.231	1.015–1.492	0.051
RV free wall LS, %	1.102	0.804–1.508	0.547
RV FAC, %	1.016	0.867–1.191	0.842
RV GLS, %	0.982	0.974–1.032	0.876
LAScd, %	0.988	0.978–1.123	0.064
LASr, %	1.120	1.010–1.242	0.031
LASct, %	1.009	0.732–1.391	0.069
RAScd, %	1.051	0.841–1.313	0.058
RASr, %	1.083	0.907–1.292	0.379

LV—left ventricle; GLS—global longitudinal strain; GCS—global circumferential strain; GRS—global radial strain; LVEF—left ventricular ejection fraction; RV—right ventricle; LS—longitudinal strain; FAC—fractional area change; LAScd—left atrial strain during conduit phase; LASr—left atrial strain during reservoir phase; LASct—left atrial strain during contraction phase; RAScd—right atrial strain during conduit phase; RASr—right atrial strain during reservoir phase.

## Data Availability

Not applicable.

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
