# Peer review of "Association of Whole-Heart Myocardial Mechanics by Transthoracic Echocardiography with Presence of Late Gadolinium Enhancement by CMR in Non-Ischemic Dilated Cardiomyopathy"

_jcm, 2022, doi:10.3390/jcm11226607_

Round 1

Reviewer 1 Report

The authors presented a study on a topic of interest to the researchers in the field of non-ischemic dilated cardiomyopathy. It demonstrated MF is related to worse of the whole-heart myocardial mechanics. Additionally, LV GLS, LASr and LV GCS are the most accurate 2D echocardiography predictive factors for presence of MF. The analysis is profound and the figures and tables given in the manuscript are suited to support and illustrate the results. To support the study results, cohorts from literature were additionally presented.

However, the report seems to have some issues, including

1.      There were some written mistakes requiring intensive grammatical editing.

2.      The title of the paper should better be more specific, including the technique that the present study used.

3.      CMR feature tracking, as a reliable technique, could quantify myocardial deformation accurately in different orientations. Why did authors not evaluate the whole-heart myocardial mechanics by CMR feature tracking in this study?

4.      In practice, the increasing signal on LGE inherent to normal pericardial fat is often difficult to differentiate from subepicardial LGE within the myocardium. How did the authors differentiate LGE from normal pericardial fat signal on LGE?

5.      Were T2 weighted images performed? This is fairly standard for cardiac MRI in patients with a suspected cardiomyopathy.

6.      LGE detection and quantification can vary with window and level adjustments. Was a specific threshold used for evaluation of LGE?

7.      The present study evaluated potential relationship between myocardial mechanics of different chambers of the heart with presence of MF and determined the accuracy of the whole-heart myocardial strain parameters to predict MF in patients with NIDCM. However, it did not represent actual prognosis/occurrence of events. Therefore, I was sceptical about the rigor of  the clinical significance. Please state limitations of your study, such as sources of potential bias, statistical uncertainty, and generalizability.

The discussion should be in a focused, logical and sound manner, and irrelevant details must be deleted.

Author Response

Dear reviewer,

We are thankful for your important remarks for us to improve our manuscript. We have tried to answer all your questions and comments.

  1. There were some written mistakes requiring intensive grammatical editing.

English was corrected.

  1. The title of the paper should better be more specific, including the technique that the present study used.

We changed our manuscript title as You recommended.

  1. CMR feature tracking, as a reliable technique, could quantify myocardial deformation accurately in different orientations. Why did authors not evaluate the whole-heart myocardial mechanics by CMR feature tracking in this study?

While we find feature tracking to be a great technique for strain analysis, we wanted this paper to be focused on ultrasound's ability to distinguish higher risk patients with NIDCM. We therefore feel that adding feature tracking analysis would not help to expand the main idea and would introduce unnecessary complication.

  1. In practice, the increasing signal on LGE inherent to normal pericardial fat is often difficult to differentiate from subepicardial LGE within the myocardium. How did the authors differentiate LGE from normal pericardial fat signal on LGE?

We fully agree that in some cases it is difficult to differentiate subepicardial LGE from pericardial fat. In applicable slices, where available, T1 mapping images were reviewed for better differentiation. Other than that, no special techniques were used and the decision was based on judgement of an experienced observer.

  1. Were T2 weighted images performed? This is fairly standard for cardiac MRI in patients with a suspected cardiomyopathy.

Per our standard protocol in patients with suspected DCM, T2 images were performed in every patient. In some patients that were scanned after introduction of multiparametric mapping in our institution, T2 mapping images were also peformed.

  1. LGE detection and quantification can vary with window and level adjustments. Was a specific threshold used for evaluation of LGE?

Since we only performed qualitative analysis for presence/absence of LGE, no specific thresholds were used to differentiate LGE from potential healthy myocardium and the decision was based on the judgement of an experienced observer.

  1. The present study evaluated potential relationship between myocardial mechanics of different chambers of the heart with presence of MF and determined the accuracy of the whole-heart myocardial strain parameters to predict MF in patients with NIDCM. However, it did not represent actual prognosis/occurrence of events. Therefore, I was sceptical about the rigor of the clinical significance. Please state limitations of your study, such as sources of potential bias, statistical uncertainty, and generalizability.

We agree that these our results didn’t represent prognosis of follow-up events. However, the follow-up of our patients is predicted after one year after including to the study.  This study wasn’t focused on prognosis of events, it’s expected in our future articles. As You mentioned, we wanted to evaluate the relationships between mechanical properties of all cardiac chambers as assessed by STE and presence of LGE on CMR imaging in a prospective cohort of NIDCM patients. These results could be useful to testing their prognostic value for future events.

Limitations were stated. Thank You for this remark.

  1. The discussion should be in a focused, logical and sound manner, and irrelevant details must be deleted.

The discussion section was corrected as recommended. Irrelevant details such as VT relation with MF were deleted.

Reviewer 2 Report

Dear authors,

I congratulate you for your work with this study. It is an interesting study evaluating the best parameters for fibrosis prediction with echocardiography. Still, I have some suggestions to make:

1.    The main results of this manuscript come from this sentence: “Logistic regression analysis showed that LV GLS, LV GCS, and LASr were independently associated with presence of MF”. Was this a multivariable analysis? If so, there is no information of that in the methods part of the abstract, and the results should be presented in a Table with the other variables that were analysed in this multivariable analysis. For example: Was LVEF also independently associated with myocardial fibrosis?

2.    The introduction part of the manuscript has the following sentence: “Advanced echocardiographic techniques, such as, speckle tracking echocardiography (STE) allow detection of structural cardiac abnormalities beyond LVEF and have been shown to correlate with presence of MF in various clinical scenarios, data in NIDCM is lacking.” This suggest that this manuscript is the first study evaluating this correlation. However, this is not true. Other studies have previously evaluated this correlation (examples:

JCF 2021; Wang J. et al., Assessment of Myocardial Fibrosis Using Two-Dimensional and Three-Dimensional Speckle Tracking Echocardiography in Dilated Cardiomyopathy With Advanced Heart Failure;

Circulation 2021 Xie M. et al., Biventricular Myocardial Strain Correlates With Myocardial Fibrosis in Patients With End-stage Dilated Cardiomyopathy: A Study Using Three-dimensional Speckle Tracking Echocardiography)

Your work also describe this fact in the discussion section of the manuscript: “so using GLS with speckle tracking echocardiography is more sensitive in finding early LV dysfunction, and had the strongest accuracy for detecting LV fibrosis [22].”

This part of the introduction should be modified accordingly.

3.    “We did not find an association between the clinical data of patients such as age and gender and presence of MF. However, our results show that more ventricular tachycardia or fibrillation episodes were detected in patients with MF and these findings confirm the fact that cardiac fibrogenesis is associated with major arrhythmic events. This association could be explained higher ventricular stiffness index and more impaired ventriculoarterial coupling compared with those without MF [18]. Di Marco et al. reported the relation between MF and ventricular arrhythmias irrespective of LVEF [19]. Thus, the analysis of presence of the MF could improve patients’ selection for implantable cardioverter-defibrillator (ICD) in primary prevention of SCD. The data are lacking on the influence of gender or age on the prevalence and extent of LGE in NIDCM. Some authors have reported that only male sex predicts the presence of MF [16], and these findings are confirmed in patients with NIDCM [15, 20]. In our study group there were statistically significantly more men (men vs women: 71 (64%) and 30 (27%), respectively), but there was no difference in the presence of MF.”

I think that the relationship between GLS and VT are well described but is not the focus of your study. I suggest that the discussion part focus on the relationship between echo parameters and fibrosis prediction by LGE.

4.    There are no limitations of the study described. This section needs to be added.

5.    Regarding LGE I have two questions:

-      Was right ventricular LGE evaluated in the Cardiac MRI? If not this could explain the results regarding RV and RA strain.

-      You analyse the results regarding having no fibrosis versus having fibrosis. However, 20% of fibrosis could be different from only 2% of fibrosis. Do you have this quantification of LGE?

Minor suggestions:

1) English needs some revision: 

Some examples:

Conclusions sections of the abstract: " in NIDCM patients, MF is related with worse of the whole-heart myocardial mechanics."

Abstract section: "All patients received invasive coronary angiography"

2) This sentence from the results part of the manuscript should be in the methods part: "The cut‐off value of the predictive model was defined as the point that yielded the maximum value of the sum of sensitivity and specificity."

Author Response

Dear reviewer,

We are thankful for your important remarks for us to improve our manuscript. We have tried to answer all your questions and comments.

  1. The main results of this manuscript come from this sentence: “Logistic regression analysis showed that LV GLS, LV GCS, and LASr were independently associated with presence of MF”. Was this a multivariable analysis? If so, there is no information of that in the methods part of the abstract, and the results should be presented in a Table with the other variables that were analysed in this multivariable analysis. For example: Was LVEF also independently associated with myocardial fibrosis?

Binary Logistic Regression Model in our manuscript was used. In general, this model was employed to model the outcomes of a categorical dependent variable (MF at 0 and 1 level; ‘0’ for not having MF and ‘1’ for having MF). The table of binary logistic regression analysis was supplemented with the other variables.

  1. The introduction part of the manuscript has the following sentence: “Advanced echocardiographic techniques, such as, speckle tracking echocardiography (STE) allow detection of structural cardiac abnormalities beyond LVEF and have been shown to correlate with presence of MF in various clinical scenarios, data in NIDCM is lacking.” This suggest that this manuscript is the first study evaluating this correlation. However, this is not true. Other studies have previously evaluated this correlation (examples:

JCF 2021; Wang J. et al., Assessment of Myocardial Fibrosis Using Two-Dimensional and Three-Dimensional Speckle Tracking Echocardiography in Dilated Cardiomyopathy With Advanced Heart Failure;

Circulation 2021 Xie M. et al., Biventricular Myocardial Strain Correlates With Myocardial Fibrosis in Patients With End-stage Dilated Cardiomyopathy: A Study Using Three-dimensional Speckle Tracking Echocardiography)

Your work also describe this fact in the discussion section of the manuscript: “so using GLS with speckle tracking echocardiography is more sensitive in finding early LV dysfunction, and had the strongest accuracy for detecting LV fibrosis [22].”

This part of the introduction should be modified accordingly.

The introduction and discussion section were modified.

  1. “We did not find an association between the clinical data of patients such as age and gender and presence of MF. However, our results show that more ventricular tachycardia or fibrillation episodes were detected in patients with MF and these findings confirm the fact that cardiac fibrogenesis is associated with major arrhythmic events. This association could be explained higher ventricular stiffness index and more impaired ventriculoarterial coupling compared with those without MF [18]. Di Marco et al. reported the relation between MF and ventricular arrhythmias irrespective of LVEF [19]. Thus, the analysis of presence of the MF could improve patients’ selection for implantable cardioverter-defibrillator (ICD) in primary prevention of SCD. The data are lacking on the influence of gender or age on the prevalence and extent of LGE in NIDCM. Some authors have reported that only male sex predicts the presence of MF [16], and these findings are confirmed in patients with NIDCM [15, 20]. In our study group there were statistically significantly more men (men vs women: 71 (64%) and 30 (27%), respectively), but there was no difference in the presence of MF.”

I think that the relationship between GLS and VT are well described but is not the focus of your study. I suggest that the discussion part focus on the relationship between echo parameters and fibrosis prediction by LGE.

The discussion section was corrected as recommended.

  1. There are no limitations of the study described. This section needs to be added.

             Limitations were added.

  1. Regarding LGE I have two questions:

    -      Was right ventricular LGE evaluated in the Cardiac MRI? If not this could explain the results regarding RV and RA strain.

In patients with NIDCM the RV myocardium is usually too thin to reliably differentiate small patches of LGE from trabeculae or subepicardial fat. Therefore the analysis was mainly focused on LV and insertion points of the RV. There were no patients with extensive RV LGE included in this study, since it is usually a feature of other specific diseases, such as ARVC, ischaemic cardiomyopathy or sarcoidosis. 

2

-      You analyse the results regarding having no fibrosis versus having fibrosis. However, 20% of fibrosis could be different from only 2% of fibrosis. Do you have this quantification of LGE?

We agree that higher amounts of fibrosis is associated with worse prognosis, however, we did not quantify the extent of LGE in this cohort. We felt that the cohort was too small for  further differentiation to yield results from which reliable conclusions could be drawn.

Minor suggestions:

1) English needs some revision: 

Some examples:

Conclusions sections of the abstract: " in NIDCM patients, MF is related with worse of the whole-heart myocardial mechanics."

Abstract section: "All patients received invasive coronary angiography"

English was revised.

2) This sentence from the results part of the manuscript should be in the methods part: "The cut‐off value of the predictive model was defined as the point that yielded the maximum value of the sum of sensitivity and specificity."

It was corrected.

Round 2

Reviewer 2 Report

The limitation of not being able to quantify the extent of LGE should be added to the limitations section of the manuscript.

English still needs some minor revision.

Author Response

Dear reviewer,

thank you once again for your important remarks.

1. The limitation of not being able to quantify the extent of LGE should be added to the limitations section of the manuscript.

It was added.

2. English still needs some minor revision.

English revision was done accurately.